# What can machines teach us in our journey of reproducing human scientific creativity?

### Abstract

In the race toward creating a strong AI, we have historically focused on replicating human intelligence. For many advanced tasks such as language and image generation, complex classifications in fields such as medicine, computer vision and other sensor data in self-driving cars, we have been successful. However, for complex behaviours like creativity, we often deem machines incapable. Maybe we are desperate to have something of our own, that machines could never do. Maybe we are too prideful in our own intelligence. What if we were tasked to build a truly creative AI capable of intuition and insight? What should we consider? Would replicating human abilities be the best option, or could we make something even better?

This article holds a mirror up to us and explores scientific creativity. We first explore the many properties that may allow machines to surpass humans in creative insight, such as unbounded effort and lack of competition. We should exploit these, rather than limit them in the attempt to make AI more 'human-like'. In the second half of this article, we realise there are many traits we have overlooked in ourselves, that we should strive to emulate in machines. There is no doubt that machines someday could mimic human creativity. The purpose of this reflection is to realise it is not about what we can build, but what we should build.

## Introduction

In the race toward creating a strong AI, we have historically been focused on understanding and replicating human intelligence in machines. For complex behaviours like creativity, we deem machines incapable. *Maybe we are too prideful in our own intelligence*. A true general AI should capture the foundations of intelligence rather than be a carbon copy of the human mind. This article holds up a mirror to human creativity, encourages critical thinking about areas where machines excel over humans, as well as qualities we have overlooked in ourselves. Through this reflection, we realise that it is not what we can build, but what we should build.

We first explore properties allowing machines to surpass humans, such as unbounded effort and lack of competition. In the second half of this article, we compare these to qualities which directly benefit human creativity (that we should strive to emulate in machines).

Imagine the brilliance of Albert Einstein's mind, a brain so unique that, when examined, it revealed an extraordinary network of connections unlike any other. This intricate web of neurons may have been the secret behind his unparalleled ability to revolutionise physics. Could a machine possess a similar spark? Not just calculating but creating, not just computing, but conceiving ideas that challenge the very fabric of our understanding. Can AI ever achieve this level of creativity, or are we chasing a mirage in the digital desert? To critically explore this topic, it is essential to first understand what creativity means in humans, especially in comparison to machines.

## Creativity

As Steve Jobs famously remarked, "Creativity is just connecting things". In humans, scientific creativity is often characterised by the ability to produce ideas or solutions that are both original and valuable. It involves making unexpected connections, thinking divergently, and applying insights in innovative ways (Roberts et al. 2021).

Unlike intelligence, which focuses on logical reasoning and problem-solving, creativity requires exploring new possibilities and recognizing patterns that are not immediately apparent. The central question in developing creative machines is whether machines can achieve similar creative capabilities. Unlike humans, AI systems leverage vast datasets and advanced algorithms to detect patterns and make connections at scales beyond human capability. Through deep learning, neural networks, and other machine learning techniques, machines are believed to replicate certain aspects of human creative processes.

However, creativity, according to philosophers, can be mainly summed up into two forms: "Big-C" creativity, referring to groundbreaking, paradigm-shifting innovations such as the theory of relativity or the invention of the Internet, and "little-c" creativity, which refers to everyday problem-solving and creative expression (Gilhooly and Gilhooly 2021). Understanding these distinctions is crucial when evaluating the creative potential of machines.

This brings us to an important question: What type of scientific creativity do our current AI systems exhibit? Consider the creativity displayed by DeepMind's AlphaGo. It surprised even the most skilled human players with unexpected strategic moves in the game of Go. particularly Move 37. Was that intentional? Were these moves preprogrammed? Obviously not; they emerged from AlphaGo's

ability to explore a vast array of possibilities using deep neural networks and Monte Carlo Tree Search algorithms. This kind of innovation prompts us to ask whether such actions represent true creativity or merely reflect the execution of advanced computational strategies.

Currently, AI systems are primarily designed to achieve "little-c" creativity by automating routine creative tasks, such as generating music, writing text, or creating visual art. While AI has demonstrated remarkable capabilities in "little-c" creativity, achieving "Big-C" creativity would require machines to move beyond imitation and produce work that fundamentally alters our understanding of a field (Gilhooly and Gilhooly 2021). This remains an elusive goal, raising questions about the true nature of machine creativity. However, even within these limitations, machines wield a different kind of creative power, one that does not mimic human thought, but thrives in areas where our minds often falter.

## Advantages that machines have over humans

The first advantage of machines is their lack of emotion. Emotions can sometimes cloud human judgement. We often develop attachment to our ideas. This motivates ownership and leads to their further development, but also limits further ideation of alternative approaches. Contrastingly, machines have unbiased evaluation of ideas; they are not limited by their first good idea and can choose the best one when finalising the solution.

Detachment also makes machines free from competition, allowing open sharing of findings. History is littered with examples of competition hindering progress: for example, the bitter rivalry which led to Newton-Leibniz calculus controversy, and the race between Elisha Gray and Alexander Graham Bell to invent the telephone. Awards such as Nobel prizes incentivise scientific discovery, but also accentuate competition and a "hero complex".

Similar to immunity from emotion, machines are also free from self-doubt. Human effort into a task is dictated by perception of success. We create a ceiling for ourselves in what we believe we can achieve. This often translates to our actual achievements, but does not speak to each person's potential intelligence or creativity. Our education, parents, peers, where we live, gender, and ability/disability all contribute to who we are, and more importantly what we think we can do. This suggests that creativity and success in a field is driven more by opportunity and circumstance, rather than innate intelligence.

Machines are free from worry of how others perceive them, or even how they perceive themselves, meaning their progress is not hindered by comparison and self-doubt. The reclusive mathematician Grigori Perelman, after proving a very important conjecture in mathematics, rejected the highest award in mathematics and a prize award of a million dollars. He said that the awards were irrelevant for him and that if the proof is correct then no other recognition is needed. In a world where we hero worship Nobel laureates and glorify external achievements and awards, Perelman had completely rejected the idea of external validation. He was fulfilled by the joy of working on problems and not by any external reward at the end of the journey: a life lived on his own terms. Can we ever replicate this in machines?

Circumstance also dictates a person's quality of education. In both humans and machines, high quality foundational knowledge (training data) is key to high creativity. Children who have a good education are more likely to do well in life. There is a parallel to how high quality training data can unlock a computer model's full potential. The two things that contribute to a computer model's success are: its architecture (which gives it the ability to learn), and its training data (which it learns from). Nowadays, with the availability of such complex neural architectures, it is often the training data that limits the model. Hence, it is difficult for us to even know the model's true capability.

Machines seemingly have unbounded effort: they do not tire. This allows them to practise the creative process. The first question this raises is, can creativity be practised? Scientific creativity seems to manifest as 'eureka' moments, but how much can we trust this? We only remember what gets stored in our brain; perhaps it is only those emotional breakthroughs that are recorded. We ignore the less glamorous computation leading up to it (Hadamard 1945). In reality, the creative process might be something that can be improved over time, as our brain gets better at making the connections that lead to those 'eureka' moments.

Jacques Hadamard, in his seminal book "The psychology of invention in the mathematical field" (Hadamard 1945), spoke about the role of the subconscious in solving difficult problems: he related the story of the French polymath Henri Poincare struggling with a mathematical problem for years. One day while visiting a friend he got on a bus: as soon as he got on the footboard, the solution came to him. In reality, Poincare was thinking about the problem (incubating) for a long time (he only remembered the idea coming to him when he got on the bus).

Pat Langley considers memory as a graph (Langley and Jones 1997), with nodes containing objects (or experiences) and edges of different weights representing the strength of connections between them. A creative insight is an analogical mapping between a node with a familiar known process, and another with a presently unknown process. This mapping is discovered as activation spreads from the new unfamiliar activated node, outward to neighbouring nodes, proportional to the edge weights. If activation reaches a node above a threshold value, an analogy is formed.

We could liken "practising creativity" to strengthening connections between nodes, building a more comprehensive mental model of the interactions in the physical world. This in turn allows us to make better analogies. By adjusting the reward function, we can even train machines (or ourselves) to make more abstract connections between nodes further apart in the memory space during training, allowing for seemingly 'more creative' insights. Treating creativity as an innate ability rather than one that can be practised, is hindering our progress in creating an AI in this field. Perhaps we need a more humble approach to artificial creativity.

A machine's unbounded effort also allows it to generate more creative insights than humans who have limited capac-

ity. Langley (Langley and Jones 1997) details how all generated insights need verification. Some may be false positives, which need to be discarded. We can set machine parameters to generate a much greater number of insights. A larger proportion may be false positives, but the overall number of true positives will be greater too. Either the machine could filter these for the correct solution, or a human could (leveraging the power of human-machine collaboration, where both human and machine complement each other).

The above arguments highlighted advantages machines have over humans. However, many human traits such as collaboration and motivation contribute to creative outputs, which have not yet been considered. These are qualities we can try to emulate in machines.

## Advantages that humans have over machines

Collaboration is a relatively underexplored phenomenon fueling creativity. Humans rely on collaboration for inspiration and verification (through peer review). The innovations built at societal level are much greater than what could be achieved by an individual.

Machines could benefit from such a collaborative paradigm. Two machines with the same architecture but different random starting initialisations could arrive at two different theories. Sharing these insights could prove invaluable. Zhuge et al. (Zhuge et al. 2023) discuss the effectiveness of AI societies of mind for natural language, in tasks such as question answering and image captioning. The neural networks solve problems in a 'mindstorm' interviewing each other in a virtual society of sorts. For humans, the act of explaining to others often solidifies ideas in our minds. Similarly, for machines, the act of communicating information with other machines in natural language might reinforce the mental model, by reinforcing relevant paths in the graph representation of memory discussed earlier.

Motivation drives creativity in humans. Earlier we recognised a lack of competitiveness as an advantage for machines. However, competition is also a great motivator. Competition for recognition fuels hype cycles, which help us decide where to direct our efforts; for example, rapid innovation is currently happening in generative AI. Hype cycles could be beneficial for machines. Multiple machines exploring the solution space increases the likelihood of a good solution. If those machines all arrive at the same solution, it gives good verification to the group that this path should be explored further. More machines also result in more information sharing and iterative building of solutions.

Humans have a major tool at our disposal that machines are playing catch up on: our bodies. Embodied intelligence (as presented by Hubert Dreyfus (Dreyfus 1967) and philosophers such as Edmund Husserl and Maurice Merleau-Ponty) is the idea that an AI would need a physical body to be truly intelligent. In the case of scientific insight, we humans are able to experiment ourselves with potential hypotheses from when we are babies. This helps us build a strong mental model of the physical world. One could argue we need to have this deeper understanding of our environment since we are required to operate our body within it: a problem which an AI living in a computer does not face.

Finally, there are perceived weaknesses that actually play to our strengths, such as forgetting and laziness. In Langley's representation of memory (Langley and Jones 1997), forming an analogy involved activation spreading outwards from the triggered unknown concept to neighbouring nodes. In this representation, more domain knowledge actually means an analogy is less likely, as the activation signal is divided amongst many different neighbours. This means the signal is less likely to be above a threshold value when it reaches the node with the analogy. This can be observed in real life, where a fresh set of eyes can help solve a problem. Forgetting, therefore, would mean some of these neighbouring connections fade over time, effectively pruning the space for only the useful information. Forgetting can also be useful in cases of concept drift, where the ground truth can change over time.

Laziness can lead to innovation. Previously we argued that unbounded effort allows machines to discover many more analogies, and as a result more true positives. However, implementing 'creative' machines on a wide scale will be resource constrained. Finding the quickest path to success may yield more creative solutions than unlimited resources; for example, the webcam was invented by scientists in Cambridge University to monitor when their coffee pot was empty! Bill Gates famously said that companies should hire a lazy person because they will find the most efficient solution to the problem.

Dreams are also an important mechanism in creativity. Hadamard in his book (Hadamard 1945) gives the example of August Kekule who famously was meditating on the structure of the benzene molecule. While thinking about this problem, Kekule fell asleep. In his dream, he saw a picture of a snake trying to eat its own tail. Upon waking up, he realised that the structure of the benzene molecule can be a circular ring of carbon atoms. A future AI that attempts to tackle scientific creativity should also have a mechanism for artificial dreams that allows it to connect pieces of information and concepts that would have been difficult to connect otherwise.

The prolific mathematician Srinivasa Ramanujan frequently attributed his discoveries to dreams that he had. His creativity, he claimed, came from his dreams. However, he could not explain how he came up with the mathematical expressions. Explainability may be important in future creative machines.

## Discussion

How do we go about building a truly creative AI? The key is balance. There are many parts of our brains we should emulate (such as collaboration, motivation, forgetting, laziness and dreams). However, we should also preserve advantages machines have over humans (such as lack of emotion, competition, lack of self-doubt and unbounded effort), respecting our own limitations. *What we create requires deep thought, rather than blind replication of human abilities.*

We often have strict benchmarking processes for machines on tasks that are designed for humans. Google's Gemini model was evaluated on human strengths: speech, maths, and common-sense reasoning. What benchmarks we

strive for now, could eventually become limiting in the future. The ultimate endeavour could be to build something better or different that complements humans. We do not need another artificial brain similar to the 8 billion human brains we already have today.

Once we achieve AI creativity, will we recognise it? The Turing Institute predicts that AI will execute Nobel-prize winning work by 2050. In reality, no matter how intelligent the AI is, we might just end up calling it a tool which supplements human intelligence. Many intelligent inventions fall victim to this; the printing press, the calculator and the computer were all revolutionary at conception, but eventually became commonplace.

Maybe AI is just an advanced tool. But what does that make us? *We tend to overvalue our own intelligence, and devalue the intelligence of AI.* OpenAI's video generation model, Sora, exhibits a deep understanding of interaction in the physical world, however it may not have the same underlying mental model as us. Does this mean it is not intelligent? There does not have to be one correct model through which we understand a process. For example, bridges and roads were built before calculus, and both classical and relativistic physics co-exist today. We cannot have it both ways. *If we are intelligent, so is AI. And if AI is just a tool, perhaps we are equally unremarkable.*

Creativity is deeply intertwined with intelligence. For a machine to be creative, it must engage in some form of thinking. But can computers think? Can they be creative? These questions might seem absurd, much like asking if trees can grow. But if computers do not think, what exactly do they do? The answer lies in understanding that computers don't think as humans do, just as trees do not grow as humans grow. These entities are distinct, governed by different natural laws. Computers do not replicate human thinking—they orient it. Their unique ways of processing information are precisely what we need them to be, why we need them, and what we need for the future. This is what we must aim to build.

We may not require consciousness in machines to achieve machine creativity. What is essential is a learning machine—one that possesses the capability to integrate experience into its decision-making processes. This concept parallels how immune systems retain memory of past infections and how human brains store and utilise memories. In other words, when AlphaGo was trained to play Go, did it rely solely on predefined algorithms? No. Instead, it learnt from extensive gameplay data, developed novel strategies, and subsequently demonstrated creativity to the extent of surprising and teaching expert players, such as Lee Sedol, new approaches in their own game.

This capacity for learning is what we need to build in machines to enable them to transcend mere determinism, evolving into a system that is adaptive, complex, and inherently creative.

## Ethics

This potential future may make us think about human redundancy and the ethical implications of such advanced intelligence.

As machines become capable of being genuinely creative, i.e. generating original work, the question of ownership becomes increasingly complex. Who owns the genuine creations of a machine—the developer, the user, or the machine itself? This conundrum challenges our existing legal frameworks and forces us to rethink our understanding of creativity and ownership in the age of AI. As we push the limits of what machines can do, we must also extend our ethical and legal considerations, ensuring that our pursuit of machine creativity does not outpace our ability to manage its implications.

## Conclusion

Ultimately the process of building creative machines is also a journey of self-discovery. It is a journey where we learn how to improve ourselves and reshape society to reward innovation more greatly than we do today.

However, it is also a journey where we may discover we are not so special after all. *Creativity and intelligence may exist in different forms in machines and humans.*

We may not want to face the fact that learning and creativity can happen easily in machines and humans given enough time and data. We are only human after all!

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
