# OpenReview forum: "What can machines teach us in our journey of reproducing human scientific creativity?"
_AAAI.org/2025/Workshop/NeurMAD — AAAI 2025 Workshop NeurMAD Submission_

### Official Review · Reviewer_CzZY · 2024-12-18
**paper review**

**Rating:** 7
**Confidence:** 3

**Review:**

## Summary
The paper provides an introspective exploration of the potential for machines to exhibit scientific creativity, comparing and contrasting machine and human capabilities.
It highlights machine advantages such as unbounded effort, immunity to bias, and lack of emotional interference while recognizing human strengths like collaboration, embodied intelligence, and motivation.

The paper proposes a balanced approach to developing creative AI, urging us to consider not just what can be built, but what should be built. Through philosophical, technical, and ethical discussions, it challenges existing benchmarks and notions of creativity while envisioning a complementary role for AI in advancing human knowledge.

## Strengths
1. The writing style is engaging, using compelling analogies and historical examples like Henri Poincaré’s "bus moment".

2. The paper takes a comprehensive approach, combining insights from psychology, philosophy, neuroscience, and AI research. It discusses "Big-C" and "little-c" creativity.

3. The article offers a balanced examination of machine and human capabilities. It highlights areas where machines can surpass humans (e.g., lack of emotional attachment) while acknowledging the importance of human traits like collaboration.

4. The paper emphasizes the importance of creating AI systems that complement rather than replicate human intelligence, providing a clear and meaningful purpose for AI in scientific discovery.

---

### Official Review · Reviewer_oxZB · 2024-12-20
**neurmad paper review**

**Rating:** 6
**Confidence:** 3

**Review:**

# Summary

This paper explores the potential of artificial intelligence to replicate or surpass human creativity. It identifies advantages machines possess, such as unbounded effort, lack of emotional bias, and detachment from competition, and contrasts these with uniquely human traits like embodied intelligence, collaboration, and intrinsic motivation. The paper argues that instead of merely mimicking human creativity, AI should be designed to exploit its unique strengths for innovation rather than replication. The paper also briefly discusses ethical considerations, including the implications of machine creativity on ownership and societal roles.

# Strengths

The paper offers a compelling conceptual framework for understanding machine creativity through the lens of both cognitive science and computational advantages. It effectively highlights the strengths of machines in areas where humans are limited, such as scalability and freedom from emotional interference. The discussion around leveraging these machine-specific traits is technically grounded, particularly in ideas like memory graph representations for creative insights and reward function optimization for analogical reasoning. Additionally, its critique of human limitations, such as biases introduced by competition or self-doubt, is well-argued and relevant to AI system design.

# Weaknesses

While the paper presents a strong theoretical narrative, it lacks empirical evidence to validate its claims. The proposed ideas, such as “artificial dreams” or AI societies of mind, remain speculative without detailed implementation strategies. The discussion on creativity metrics is insufficient, failing to address how “Big-C” creativity (paradigm-shifting innovations) could be systematically identified or evaluated in machines. Furthermore, the ethical implications, while noted, are underdeveloped, leaving key questions of accountability and societal impact unresolved.

Overall, the paper provides valuable insights and raises thought-provoking questions about AI creativity. However, its theoretical nature and limited attention to practical methodologies and metrics reduce its overall technical contribution. Addressing these gaps could significantly enhance its impact.

---

### Decision · Program_Chairs · 2024-12-30

**Decision:**

Reject

**Comment:**

This is a nice essay, but it lacks math to appear at this workshop.